# OpenReview forum: "Incoherent Beliefs & Inconsistent Actions In Language Models"
_ICLR.cc/2026/Conference — Submitted to ICLR 2026_

### Official Review · Reviewer_G83G · 2025-10-30

**Soundness:** 1
**Presentation:** 1
**Contribution:** 2
**Rating:** 2
**Confidence:** 5

**Summary:**

This paper explores the consistency of “beliefs” in LLMs through a number of different experiments. First, model classification probabilities are elicited in response to limited evidence (a medical diagnosis task), then elicited again with additional evidence, in order to test how models update on new evidence. Do they update according to Bayes Rule? Apparently not. Second, it is examined whether models would place bets that relate properly to their elicited beliefs (event probabilities). Do they bet rationally? It seems not. Lastly, an experiment is conducted where a simulated user pushes back on a model’s answer to a question, and it is measured whether the model flips its answer — this flipping behavior is correlated against the model’s initial credence in its answer. There seems to be a mildly positive correlation between strength of initial belief and the model’s propensity to stick with its original answer in the face of a user challenge. Finally, the paper seeks to understand whether the three notions of belief consistency above can be understood in terms of the model’s calibration or accuracy on the underlying task. Results here are mixed.

**Strengths:**

- Very important: The paper tackles an important and timely question: do models have consistent internal beliefs that guide behavior, including self-reports and actions subject to constraints of rationality, like betting? Overall, the paper suggests that they generally do not.
- Important: The paper approaches its central question through multiple experiments that all provide some evidence for the full picture. This is necessary for a complex question like the question of whether models coherently update their beliefs and coherently act on them.

**Weaknesses:**

- Very important: I think the primary weakness of the paper is that it spreads itself too thin across too many experimental directions. By my count, the paper conducts at least five quite distinct experiments, (Secs 4, 5, 6, 6.2, and 7). This ambition immediately becomes a stumbling block for the analysis. Beginning with Section 4, I had many concerns. How could we make a general claim about Bayesian updating in models based on only one dataset? This dataset concerns a specific task, and the results could be confounded by at least three variables: (1) the task involves significant prior knowledge, (2) the task involves numerical reasoning, and (3) the task is often refused by the model when probing involves free-text generations. Without exploring the space of tasks more thoroughly, I find it extremely difficult to be confident in the claim that models do not do Bayesian updates on new evidence. Limited space also makes it hard to present all the critical details to each experiment. On Sec 3, I do not understand how p(E|.) is computed. Should we take p2 or p2* as the model’s “chief” way of computing its posterior? What should we make of them being inconsistent? Is the only conclusion that the model doesn’t do Bayesian inference, or could we conclude that it can do Bayesian inference through the computations in p2, but for some reason the computations involved in p2* are not successful and therefore don’t represent the model’s true posterior? I would go on with similar issues for each of the following sections. While each direction is very interesting, it is hard to walk away confident in the conclusions presented (and the results are often quite noisy / unstable, as in Sec. 7).
- Of some importance: Just as a technical point, there is an ambiguity in the claim at lns 317-319. It is true that a model should be more willing to defend answers that it has higher confidence in. That is not necessarily the same thing as a model’s p(answer|question). The reason for this is that the important quantity here is “stickiness” of p(answer|question), not the absolute magnitude of p(answer|question). Stickiness is a function of the strength of evidence for the answer. I find it hard to put this succinctly, but will try here. Consider being asked to bet on an event X occurring. This should depend on your p(X). If you do a little bit of research, your p(X) might be 0.7, but you might still be quite open to adjusting that number in the face of new evidence. If you’ve done tons of research, you might be quite confident that 0.7 is the right probability for the event, and you would be less willing to change your credence in the face of new evidence. The deference experiment the paper conducts should really rely on the belief stickiness, not the credence itself. The issue of course is that we don’t know how to measure belief stickiness, because we don’t know how models update their probabilities to begin with (the subject of Sec 4)!

**Questions:**

- Feel free to respond to any of the suggested weaknesses.
- Besides this, note that there is actually a decent amount of work now on Bayesian updating in LLMs, or at least how models respond to new evidence. See:

Talmor et al, Leap-Of-Thought: Teaching Pre-Trained Models to
Systematically Reason Over Implicit Knowledge, https://proceedings.neurips.cc/paper_files/paper/2020/file/e992111e4ab9985366e806733383bd8c-Paper.pdf

Wilie et al, Belief Revision: The Adaptability of Large Language Models Reasoning,  https://arxiv.org/pdf/2406.19764

Hase et al, Fundamental Problems With Model Editing: How Should Rational Belief Revision Work in LLMs?, https://arxiv.org/pdf/2406.19354

Qiu et al, Can Language Models Perform Implicit Bayesian Inference Over User Preference States?, https://openreview.net/pdf?id=arYXgfHAIh

---

> ### Author Response · Authors · 2025-11-21
> **Response Part 1 to Reviewer G83G**
>
> Thank you very much for your thoughtful and detailed review. We are glad that you found our paper tackles an important question.  We address your points below:
>
> W1:  Regarding task choice, the crux of our conclusion does not require the model to perform correct numerical reasoning, nor does it test the quality of the models’ prior knowledge. Our metric of consistency is entirely agnostic to that. Consider, for example, an entity which is perfectly Bayes-rational but has no knowledge about anything whatsoever; they should then return a probability of 0.5 for all the questions asked in this design. Doing so would _still_ return an exact match between the computed and elicited prior, unlike the models we tested. Similarly, there is no explicit numerical reasoning required per se; the entity is never asked to _actually_ compute its posterior from its constituent probabilities, but simply asked about each of those individually.
>
> Regarding whether p2 or p2* is the ‘target’, as we state above, our design is completely agnostic to this. The important relationship for consistency measurement is how close these are to each other; not the absolute value either of them take. To be very clear about this, our design does not require the model to perform _any_ computation in order to qualify as being consistent. All that is required is that when we enquire about the models’ confidences of each sub-component of the equation in lines 206-208, that when we then _ourselves_ plug this into the formula, that they match with p2.
>
> p2 not matching p2* indicates that model beliefs are not self-consistent. In general, it would be difficult to ascertain whether this is because the model fails to update its prior given the new evidence correctly (i.e. p2 is more at fault) or whether it is the case that the subcomponents constituting p2* are mis-estimated. However, in Figure 4, we do report the predictive performance of p2 and p2* and find that the elicited posterior, p2, is actually worse on the task than p2* for nearly all models. Therefore, we conclude that it is more the case of the former than the latter; but it is not possible to give an exact ‘contribution’ percentage of each effect.
>
> We do acknowledge that there may be some effect of the model refusal behavior on our results, and that we could have further tested a wider spread of datasets. We have now tested on an additional dataset inspired by the setup in Qiu et al. Specifically, we examined belief updates in a flight-suggestion task, where a user’s preferences are encoded as a vector of weights, and the model attempts to infer the preferred option between two flights. As with the diabetes dataset, we evaluate belief quality using Brier scores. The results for BS(p2,p2*) are reported below:
>
>
> | Model | Brier(el_post, calc_post) |
> | :--- | :--- |
> | Gemma 2 9B IT | 0.142 |
> | Llama 3.1 8B Instruct | 0.007 |
> | Mistral Small Instruct 2409 | 0.128 |
> | GPT-4o | 0.283 |
> | GPT-4o Mini | 0.304 |
>
> In addition, we computed the correlations of p₁, p₂, and p₂* with D, the ground truth, again using Brier scores. These results are provided here:
>
> | Model | Brier(D, p2) | Brier(D, p2\*) | Brier(D, p1) |
> | :--- | :--- | :--- | :--- |
> | Gemma 2 9B IT | 0.523 | 0.521 | 0.496 |
> | Llama 3.1 8B Instruct | 0.27 | 0.271 | 0.258 |
> | Mistral Small Instruct 2409 | 0.253 | 0.253 | 0.355 |
> | GPT-4o | 0.482 | 0.481 | 0.359 |
> | GPT-4o Mini | 0.474 | 0.48 | 0.47 |
>
> **In this dataset, we found no cases of model refusal.** Overall, we observe patterns consistent with those observed using the diabetes dataset. The relative behavior of the models and the gap between elicited and calculated posteriors are similar to our results on the diabetes dataset, and thus further support our initial findings.
>
> We have now added the additional experiment above to our current draft. Thank you again for your thoughtful input on these.
>
> Our response is continued further in the next comment.

---

> ### Author Response · Authors · 2025-11-21
> **Response Part 2 to Reviewer G83G**
>
> W2: Thank you for mentioning this point. We understand the thrust of your argument here, but we think that it is not quite applicable to our deference consistency experimental design.
>
> The outcome space of answer | question is a Bernoulli variable – it can only ever be True, or False. Therefore, the quantity p(answer | question) already quantifies the prior uncertainty of the model; a Bernoulli variable is only a function of a single parameter, and that is precisely specified by p(answer | question). Said another way, you cannot change the variance of the estimate without also changing the mean of the estimate.
>
> This would be quite different if the outcome space was actually a continuous variable. For example, if the model was asked to state the probability of some outcome occurring, then this would be a continuous random variable in [0, 1], and so one could meaningfully distinguish between a posterior probability that is tight around a particular value, versus one that has wide variance around that value (e.g. this would be plausible for a truncated Gaussian, say, or for a beta distribution).
>
> Q2: Thank you for bringing these papers to our attention. Regarding Willie et al, the paper focuses on the particular case of how LLMs perform in logical entailment tasks, given a sequence of contradictory predicates. Although interesting, this is fairly detached from our work, which focuses on the mismatch between confidences elicited from the model, and their behaviors.
>
> Talmor et al primarily focus on whether models can perform logical inference types such as hypernymy, meronymy, and approximate counting. As the paper is relatively older (published in 2020), they do so by fine-tuning RoBERTa on a set of facts gathered from datasets such as ConceptNet and WikiData. This paper is also, in our view, fairly unrelated to our work.
>
> The work by Qiu et al examines the extent to which LLMs update their estimations given new evidence over sequences of data, and in particular compare their performance to an optimal Bayesian updater. Therefore, we do feel this work is related to our paper. However, as we explain above, our main focus in the Bayesian section is not on the extent to which LLMs are ideally updating their beliefs in line with Bayes’ rule, but instead, the extent of their self-inconsistency between the computed posterior and elicited posterior. Moreover, our paper examines a wider range of rationally inconsistent behavior than just the Bayesian setting.
>
> The work of Hase et al is extremely interesting, but differs significantly from ours. The experimental work of this paper consists of training an LLM from scratch using a carefully developed pretraining corpus, such that the “knowledge” of the model can be precisely controlled; and then, using LoRA finetuning to apply model edits to selected facts. The methodology (using their own, smaller, trained-from scratch models, as opposed to our testing on modern state-of-the-art open and closed-source LLMs), motivation, and the larger focus (whether models update downstream posteriors after selective fact editing) are significantly different from ours. Nevertheless, we find the paper’s elucidation on the difficulties of editing model beliefs – including defining what model editing should actually mean when it is not possible to specify all possible variables, or when it contradicts with other held beliefs; the difficulty of measuring model beliefs; and the question of whether models truly have ‘beliefs’ in the first place – to be philosophically interesting, and the last question in particular is pertinent to our work.
>
> We have now added Qiu et al and Hase et al to our related works section, along with the above discussion. Thank you again for mentioning these papers.
>
> Thank you once again for your thorough review and your insightful comments. We made a significant effort to address each of your points, including substantial experiments and paper edits, and we think your feedback has improved our paper.  We would appreciate it if you would consider raising your score in light of our response. Do you have any other questions we can address?

---

> ### Comment · Reviewer_G83G · 2025-11-23
>
> Thanks for the thorough response! Let me try to comment on each point.
>
> > Consider, for example, an entity which is perfectly Bayes-rational but has no knowledge about anything whatsoever
>
> Yes this entity would display an exact match between the computed and elicited posterior. Yet we would be hesitant to draw the conclusion that they were doing Bayesian inference. I'm a little confused by another comment that I think is relevant here: "our main focus in the Bayesian section is not on the extent to which LLMs are ideally updating their beliefs in line with Bayes’ rule, but instead, the extent of their self-inconsistency." I thought one goal of the paper was to assess whether LLMS are ideally updating their beliefs. Is it only to assess the difference between the computed and elicited posterior? Why bother assessing this difference if not to study the coherence of models that perform Bayesian inference?
>
> > Similarly, there is no explicit numerical reasoning required per se
>
> Yeah sorry to clarify, I just meant the Diabetes task involves some numerical inputs, and models tend to vary widely in their ability to handle such tasks, so this could introduce some additional variance into the experiments.
>
> > We have now tested on an additional dataset inspired by the setup in Qiu et al.
>
> Great! These results are interesting to me and lend credence to the original conclusions.
>
> > W2: ... The outcome space of answer | question is a Bernoulli variable – it can only ever be True, or False. Therefore, the quantity p(answer | question) already quantifies the prior uncertainty of the model
>
> Thanks, this is a good point. Admittedly I don't have an exact sketch here, but I think the stickiness point still applies. I think it depends on how the model interprets the "challenge phrase." Does the challenge phrase introduce any evidence to the LLM? Pragmatically, it could. The user could be an expert on the topic, perhaps a topic which the LLM knows comparatively little about. Then the challenge phrase should constitute some evidence. Even being agnostic to the identity of the user, the LLM might interpret this challenge as a little bit of evidence.
>
> So it's kind of like saying, how should the estimator for the Bernoulli parameter respond to one additional observation? If one has already gathered much evidence for this parameter, the estimator should shift very little. If one has gathered little evidence, the estimator could shift quite a bit. But it depends on the number of observations for the parameter p, not p itself. What do you think about that?
>
> > Q2: Thank you for bringing these papers to our attention.
>
> Yeah your read on relevance seems fair. I mention them as examples of papers comparing LLMs to Bayesian models, which you're right only Qiu and Hase do. I still have this feeling about the paper presentation that it does seem like you want to take a view on models doing Bayesian inference, which is somewhat unlike your claim that the paper's "main focus in the Bayesian section is not on the extent to which LLMs are ideally updating their beliefs in line with Bayes’ rule, but instead, the extent of their self-inconsistency." The abstract says one of the goals of the paper is to examine "the ability of LLMs to coherently update their beliefs." This plus the emphasis placed on metrics like the Brier score make it seems like the paper does care about the absolute quality of the model's probabilities (which should relate to whether they do Bayesian inference over new evidence).
>
> --
>
> Given the new experiments, I would raise my score rating from 2 to 3 but only the options 2 and 4 are present, and I do not feel that the updated paper is merely "marginally below the acceptance threshold". I would still recommend to the authors to significantly focus in the paper on the most interesting 1-2 ideas, rather than the 4-5 currently present.

---

> ### Author Response · Authors · 2025-11-27
> **Response Part 1**
>
> Thank you for engaging in the review process, and the interesting discussion!
>
> > Yes this entity would display an exact match between the computed and elicited posterior. Yet we would be hesitant to draw the conclusion that they were doing Bayesian inference. I'm a little confused by another comment that I think is relevant here: "our main focus in the Bayesian section is not on the extent to which LLMs are ideally updating their beliefs in line with Bayes’ rule, but instead, the extent of their self-inconsistency." I thought one goal of the paper was to assess whether LLMS are ideally updating their beliefs. Is it only to assess the difference between the computed and elicited posterior? Why bother assessing this difference if not to study the coherence of models that perform Bayesian inference?
>
> You are right -- we will clarify here. In the quoted passage, we were comparing to Qiu et al who have a specific ground truth that they compare to, which is the Bayes-optimal update. In our case, in order to measure consistency, we do not directly compare the model’s answer to the Bayes-optimal update, or any other 'ground truth'. We compare only to what the update 'should be' given all of the other models' beliefs. Said another way: we are not evaluating how well the LLMs model intermediate terms such as P(E | D = 0, X) and so on, which must all necessarily be modelled correctly to perform a Bayes-optimal update (technically, it might be possible to do the right update without modelling them perfectly in some scenarios, but not in all possible scenarios).
>
> > Yeah your read on relevance seems fair. I mention them as examples of papers comparing LLMs to Bayesian models, which you're right only Qiu and Hase do. I still have this feeling about the paper presentation that it does seem like you want to take a view on models doing Bayesian inference, which is somewhat unlike your claim that the paper's "main focus in the Bayesian section is not on the extent to which LLMs are ideally updating their beliefs in line with Bayes’ rule, but instead, the extent of their self-inconsistency." The abstract says one of the goals of the paper is to examine "the ability of LLMs to coherently update their beliefs." This plus the emphasis placed on metrics like the Brier score make it seems like the paper does care about the absolute quality of the model's probabilities (which should relate to whether they do Bayesian inference over new evidence).
>
> Specifically with respect to Brier score, we point out that we only take the Brier score between the computed and elicited posterior as our metric of inconsistency, *not* the Brier to the ground truth (although we do measure Brier to the ground truth separately, to determine which of the computed and elicited posteriors are 'more wrong', as a separate point of analysis).
>
> >Thanks, this is a good point. Admittedly I don't have an exact sketch here, but I think the stickiness point still applies. I think it depends on how the model interprets the "challenge phrase." Does the challenge phrase introduce any evidence to the LLM? Pragmatically, it could. The user could be an expert on the topic, perhaps a topic which the LLM knows comparatively little about. Then the challenge phrase should constitute some evidence. Even being agnostic to the identity of the user, the LLM might interpret this challenge as a little bit of evidence. So it's kind of like saying, how should the estimator for the Bernoulli parameter respond to one additional observation? If one has already gathered much evidence for this parameter, the estimator should shift very little. If one has gathered little evidence, the estimator could shift quite a bit. But it depends on the number of observations for the parameter p, not p itself. What do you think about that?
>
> The point here is that we never ask the LLM 'what is the probability that your answer is correct', which would be exactly in line with your observation above. More concretely, the LLM could have a Beta distribution prior over the Bernoulli parameter (the most standard construction, since the Beta distribution is the conjugate prior to the Bernoulli), and the hyperparameters $\alpha, \beta$ of the Beta distribution correspond exactly to the 'prior observations' of success rates of Bernoulli flips i.e. to the 'strength of evidence'.
>
> In our case though, we simply elicit directly if the answer itself is correct. To be even more clear, if X is the Bernoulli random variable which takes 1 if the answer is correct and 0 if the answer is incorrect, and p is the Bernoulli parameter, we only ever get the LLM's P(X), or P(X | evidence), never P(p) or P(p | evidence).
>
> Our response is continued further in the next comment.

---

> > ### Author Response · Authors · 2025-11-27
> > **Response Part 2 to Reviewer G83G**
> >
> > We would also like to add that we have now performed an additional experiment to address concerns regarding the betting scenario, and whether it is representative as a setting for LLM actions. This experiment probes the relationship between models’ elicited beliefs and their actions in a tool-use setting. Concretely, we consider a scenario where the model is asked a fact-based question and has the option to call a web search tool that would return the ground truth answer but at some cost. We then measure how the probability of invoking this tool varies as a function of the model’s stated confidence.
> >
> > We conduct this analysis on the TriviaQA dataset [1], restricted to a random sample of size 400 of its “no context” subset where no supporting evidence or reading material is presented. In one interaction, we elicit the model’s confidence in its answer, using a prompt similar to the one used for Section 6 (without mentioning the possibility of using a tool at this stage). Then, in a separate interaction, we ask the model the same question while also informing it that there is a web search tool which can be invoked in cases where it is unsure of the true answer. Specifically, we append the following statement to the initial prompt:
> >
> > ```If you are not sure of the answer, instead of providing it, you may use the tool search("{TEXT TO SEARCH}") for web searches, which will give you reliable answers. Use this tool only when necessary.```
> >
> > As in the stick-rate versus confidence analysis of Section 6, we then plot the probability of a direct answer (i.e. not invoking the tool) against binned confidence levels (10 percentile-based bins) and compute Spearman’s rank correlation between the two. Note that a Spearman’s score of 1.00 would indicate ideal behavior here, where a lower confidence never corresponds to a lower probability of using the tool. Hence, this metric does not judge what the correct confidence level or threshold for a tool call should be – only whether the model applies whatever internal criterion it chooses in a consistent, monotonic way.
> >
> > Our detailed results are shown below:
> >
> > | Model                         | Spearman(confidence, p(direct answer)) | Accuracy | Average p(direct answer) |
> > |-------------------------------|----------------------------------------|----------|--------------------------|
> > | Llama 3.1 8B Instruct         | 0.081                                  | 0.742    | 0.464                    |
> > | Gemma 2 9B IT                 | 0.110                                  | 0.363    | 0.073                    |
> > | Mistral Small Instruct 2409   | 0.176                                  | 0.614    | 0.833                    |
> > | GPT-4o Mini                   | 0.285                                  | 0.817    | 0.372                    |
> > | GPT-4o                        | 0.212                                  | 0.979    | 0.980                    |
> >
> >
> > Across models, we observe behavior in this tool-use setting is generally poorly aligned with the elicited confidences. While the correlations are uniformly positive, suggesting models are at least directionally reasonable, they are still far from exhibiting a strong level of consistency. This result further supports our findings outlined in Section 5, and underscores our concerns that LLMs may exhibit substantial action-belief inconsistencies, especially in agentic or autonomous settings.
> >
> > [1] Joshi et al., TriviaQA: A Large Scale Distantly Supervised Challenge Dataset for Reading Comprehension. https://huggingface.co/datasets/mandarjoshi/trivia_qa
> >
> >
> >
> > Given the additional experiments and clarifications we have now provided, we hope you might consider revisiting your score once more, since we believe the new evidence directly addresses your core concerns. We genuinely appreciate the depth of your engagement and the constructive direction your feedback has given the paper.

---

### Official Review · Reviewer_FvrH · 2025-10-31

**Soundness:** 2
**Presentation:** 3
**Contribution:** 3
**Rating:** 4
**Confidence:** 4

**Summary:**

This paper investigates the ability of LLMs to coherently update their beliefs and examines whether the actions they take are consistent with those beliefs.
The authors find three main results:
1. LLMs are largely inconsistent in how they update their beliefs.
2. LLMs frequently take actions that contradict their stated beliefs.
3. Action–belief discrepancy is not strongly correlated with task performance.

**Strengths:**

The paper tackles an interesting and important question about LLM reasoning consistency and belief dynamics.

The experimental coverage is broad with a range of settings and analysis.

**Weaknesses:**

The experimental setup is not sufficiently robust without further justification.
- It is inherently more difficult for the model to estimate P(E | D = 1,X) (line 204) , since this inference task differs from the standard training objective of predicting the answer directly. Instead, it requires conditioning on the answer to infer which factors caused it. This discrepancy raises doubts about the accuracy of p_2^* and makes the comparison setup in Figure 3 less convincing.
- Section 5 focuses only on the betting task as an example of belief–action alignment. However, this might not be a representative action type, since models could be biased or unfamiliar with betting scenarios. Testing with different actions could strengthen the argument. Or the authors should provide a stronger justification for why betting in a market setting is an appropriate and convincing test of belief–action consistency.
- In Section 6, the authors only use the feedback message “Your answer to the initial question is incorrect” (line 323). More varied or informative feedback types could provide stronger evidence.

The description of the experimental setting lacks clarity. For example, in line 161, it is unclear how the second-round prompt is presented. Do you explicitly mention that additional information is being provided, or do you simply show X, E without any contextual hint?

**Questions:**

1. Would it be possible to test some of the setups on a larger open-source model (e.g., around 70B parameters) or a large reasoning model to examine whether the observed trends remain consistent? I understand that this would involve substantial additional effort, so it is reasonable if the authors do not have the capacity to perform such experiments.

---

> ### Author Response · Authors · 2025-11-20
> **Response to Reviewer FvrH**
>
> Thank you for your detailed and thoughtful review. We are glad that you found our paper addresses an interesting and important issue regarding LLMs, and that our experimental designs and analysis are broad in their coverage. We address your other points below:
>
> W1: In fact, as we show in Figure 4, p2* is actually a _better_ predictor on the task than p2. In other words, using P(E | D = 1,X) seems to be a better predictor, rather than estimating P(D = 1 | E, X) directly. Therefore, we believe our experimental design here to be sound.
>
> W2: Thank you for mentioning this point. Our use of the betting scenario stems from the clear notion of rational behavior in such a setting. In particular, the well known Dutch Book theorems indicate that any rational agent should be willing to assign a subjective numeric probability to different events, and sets properties on these that are necessary in order to avoid inconsistency, and ultimately, arbitrage opportunities.
>
> We are careful in our submission to ensure that models are capable of behaving coherently in this domain, using a simplified betting scenario involving a coin toss. We report the results of this in Appendix 10. In short, we find that most models are capable of performing the task perfectly in this simplified scenario in the linear utility setting; in the log utility setting, models do generally exhibit errors, but these are much smaller than those in our main experimental setting. We therefore concluded that our design is not confounded by model unfamiliarity or performance constraints, and does indeed probe betting-to-belief inconsistency.
>
> We agree that in future work, it would be interesting to extend this to a wider range of settings.
>
> Thank you again for raising the above point. We have clarified the motivation of our design in our current draft, as well as emphasized the results of Appendix 10 which provide evidence that our design is free of confounding factors.
>
> W3: We actually perform this experiment with 5 different prompts (such as “Are you sure your answer to the initial question is correct?”, “Revisit your solution to the initial question”, etc) which are given in Appendix 14. We realize that our description of this is not clear and we have rectified it in our current draft. Thank you for highlighting this point.
>
> W4: “In line 161, it is unclear how the second-round prompt is presented. Do you explicitly mention that additional information is being provided, or do you simply show X, E without any contextual hint?”
>
> Thank you for pointing this out. We do in fact explicitly indicate that the model should update its prediction based on the new information. Specifically, we include: “Additional synthetic information: [E]. Update whether this synthetic profile would be classified as diabetic” in the second round prompt.
>
> Q1: We have now run a subset of our experiments on an additional, larger and newer open-source model, Qwen3-32B. We performed experiments on deference consistency on the GPQA and GSM-Symbolic datasets, as well as the Bayesian experiments. We elicit the logit confidences for these experiments. Due to the larger size of the model, additional experiments are still in flight.
>
> #### Deference Consistency of Qwen3-32B (Logits)
>
> | Model | GPQA | GSM-Symbolic |
> | :--- | :--- | :--- |
> | Qwen3-32B | 0.952 | 0.800 |
>
> #### Brier Scores of Qwen3-32B
>
> | Model | Brier(calc. post, elic. post) | Brier(gt, prior) | Brier(gt, calc. post) | Brier(gt elic. post) |
> | :--- | :--- | :--- | :--- | :--- |
> | Qwen3-32B | 0.101 | 0.331 | 0.331 | 0.354 |
>
> Where calc post is the calculated posterior, elic post is the elicited posterior, and gt is the ground truth.
>
> Qwen3-32B obtains some of the highest deference consistency scores on both GPQA and GSM-Symbolic across all models tested. Further, it obtains the second lowest bayesian inconsistency, despite being worse at predicting the ground truth than most models. We tentatively conclude that Qwen3-32B appears to be among the most consistent models, in line with GPT4o.
>
> We will add the above results on Qwen3-32B, as well as the remaining set that are in flight, to our draft when they are completed. Thank you for mentioning this point.
>
> Thank you once again for your thorough review and your insightful comments. We made a significant effort to address each of your points, including substantial experiments and paper edits, and we think your feedback has improved our paper.  We would appreciate it if you would consider raising your score in light of our response. Do you have any other questions we can address?

---

> ### Author Response · Authors · 2025-11-27
> **Response Part 2 to Reviewer FvrH (new)**
>
> Following your feedback outlined in W2, we have now also run an additional experiment to probe the relationship between models’ elicited beliefs and their actions in a tool-use setting. Concretely, we consider a scenario where the model is asked a fact-based question and has the option to call a web search tool that would return the ground truth answer but at some cost. We then measure how the probability of invoking this tool varies as a function of the model’s stated confidence.
>
>
> We conduct this analysis on the TriviaQA dataset [1], restricted to a random sample of size 400 of its “no context” subset where no supporting evidence or reading material is presented. In one interaction, we elicit the model’s confidence in its answer, using a prompt similar to the one used for Section 6 (without mentioning the possibility of using a tool at this stage). Then, in a separate interaction, we ask the model the same question while also informing it that there is a web search tool which can be invoked in cases where it is unsure of the true answer. Specifically, we append the following statement to the initial prompt:
>
>
> ```If you are not sure of the answer, instead of providing it, you may use the tool search("{TEXT TO SEARCH}") for web searches, which will give you reliable answers. Use this tool only when necessary.```
>
>
> As in the stick-rate versus confidence analysis of Section 6, we then plot the probability of a direct answer (i.e. not invoking the tool) against binned confidence levels (10 percentile-based bins) and compute Spearman’s rank correlation between the two. Note that a Spearman’s score of 1.00 would indicate ideal behavior here, where a lower confidence never corresponds to a lower probability of using the tool. Hence, this metric does not judge what the correct confidence level or threshold for a tool call should be – only whether the model applies whatever internal criterion it chooses in a consistent, monotonic way.
>
> Our detailed results are shown below:
>
> | Model                         | Spearman(confidence, p(direct answer)) | Accuracy | Average p(direct answer) |
> |-------------------------------|----------------------------------------|----------|--------------------------|
> | Llama 3.1 8B Instruct         | 0.081                                  | 0.742    | 0.464                    |
> | Gemma 2 9B IT                 | 0.110                                  | 0.363    | 0.073                    |
> | Mistral Small Instruct 2409   | 0.176                                  | 0.614    | 0.833                    |
> | GPT-4o Mini                   | 0.285                                  | 0.817    | 0.372                    |
> | GPT-4o                        | 0.212                                  | 0.979    | 0.980                    |
>
>
> Across models, we observe behavior in this tool-use setting is generally poorly aligned with the elicited confidences. While the correlations are uniformly positive, suggesting models are at least directionally reasonable, they are still far from exhibiting a strong level of consistency. This result further supports our findings outlined in Section 5, and underscores our concerns that LLMs may exhibit substantial action-belief inconsistencies, especially in agentic or autonomous settings.
>
> [1] Joshi et al., TriviaQA: A Large Scale Distantly Supervised Challenge Dataset for Reading Comprehension. https://huggingface.co/datasets/mandarjoshi/trivia_qa
>
> Thank you once again for your thoughtful feedback, and we have made a significant effort to address your points. We would greatly appreciate it if you could please consider increasing your score accordingly. Do you have any further questions?

---

### Official Review · Reviewer_WZDe · 2025-11-01

**Soundness:** 2
**Presentation:** 3
**Contribution:** 2
**Rating:** 4
**Confidence:** 5

**Summary:**

This paper presents a comprehensive empirical investigation into the behavioral consistency of Large Language Models (LLMs) in sequential and interactive scenarios. The authors probe two core aspects: 1) the coherence of belief updates upon receiving new evidence, measured against Bayesian norms, and 2) the alignment between an LLM's stated confidence and its subsequent actions (betting in a prediction market and defending answers under challenge). Using datasets like Pima Indians Diabetes and Metaculus, the study finds significant apparent inconsistencies. Models deviate substantially from Bayesian updating, and their posterior beliefs often degrade predictive accuracy compared to their priors. Crucially, models frequently place bets that are directionally opposite to their elicited confidences and exhibit only moderate "deference-consistency" (i.e., defending high-confidence answers more robustly). The paper also explores mitigation via prompting and activation steering. A central, unresolved question underpinning the entire work is the validity of the elicited confidences as a true proxy for the model's "beliefs."

**Strengths:**

1. Research Topic: Understanding LLM behavior beyond static benchmarks is crucial for deployment.

2. Novel Experimental Paradigms: The betting market and deference consistency experiments are creative approaches to studying LLM behavior.

3. Interesting observation: The finding of a negative correlation between static calibration (ECE) and consistent agentic behavior (betting) is the paper's most significant contribution.

**Weaknesses:**

1. Related Work - Confidence slicitation and calibration: expecting few more works such as https://arxiv.org/abs/2508.15260 and https://arxiv.org/abs/2503.22353. The second one also focuses on the LLM consistency problem.

2. The entire analytical framework is built on the premise that elicited confidences are a valid measure of an LLM's "belief." The paper does not sufficiently validate this premise or engage deeply with the philosophical and practical challenges of defining and measuring belief in a stateless, auto-regressive model. The observed phenomena might be better described as "inconsistencies between different output behaviors" rather than between "belief and action."

3. Regarding the Bayesian exp, it's kinda circular reasoning to me. Again, elicited confidences are valid is the assumption. If inputs are unreliable, p2 is meaningless. You're comparing unreliable generations to each other, not testing coherence.

4. The studied actions are limited to betting and textual deference. The generalizability of these findings to other action spaces (e.g., tool use, physical reasoning) remains an open question.

**Questions:**

1. To what degree do the confidence estimates from your three elicitation methods (logits, sampling, verbal) correlate with each other for the same question/model? Low convergence would suggest we are not reliably measuring a stable underlying construct, which would critically challenge the interpretation of your results.

2. Any reference or evidence supports the statement that the elicitation methods are reliable? It could be cited or proved. But I did not find it. A simple alternative story could be: Elicitation methods are unreliable; behavioral measures (betting) may be more accurate.

---

> ### Author Response · Authors · 2025-11-20
> **Response Part 1 to Reviewer WZDe**
>
> Thank you very much for your detailed review. We are glad that you found our paper addresses a crucial issue for LLM deployment, that our experimental designs are novel and creative, and that the negative correlation between calibration and action-consistency is a significant finding and contribution.
>
> We address your other points below:
>
> W1: Thank you for mentioning these interesting papers! For the first, ‘Deep Think With Confidence’, we note that the date it was submitted to arXiv was less than a month before the current conference’s submission deadline, and so it constitutes concurrent work. The second paper, by Li et al, is indeed relevant to our work. It also addresses LLM deference, and in particular examines the extent of accuracy degradation of LLMs over repeated interactions with varying types of adversarial prompting methods. We particularly appreciate the extension to multiple rounds of interaction. However, although the paper mentions ‘consistency’ in the title, their work focuses largely on accuracy. Our work has an entirely different focus – we do not examine accuracy degradation, but instead, the consistency of model behavior with respect to defending their original answers. Although Li et al do also measure confidence, they do not relate it directly to the probability of the LLM changing its answer. In addition to testing on a diverse range of datasets across a wide range of open and closed-source models, and introducing a metric to measure deference-consistency, this only constitutes 1 of 3 main parts of our paper.
>
> We have now incorporated the above into our Related Work section. Thank you again, for bringing these papers to our attention.
>
> W2: We agree that we could have been more detailed in our exposition regarding the difficulty of ascertaining the ‘true beliefs’ of LLMs. The field as a whole has not yet come to a single conclusive metric of the above; as you point out, even defining the concept clearly remains an unresolved problem. For example, [Hase et al](https://arxiv.org/pdf/2406.19354) enumerate some of the difficulties inherent in editing model beliefs, including whether they have a single such set of beliefs, or their beliefs change depending on the ‘agent persona’ they adopt.  As there is no consensus on this, we test all the 3 main methodologies in common use for belief-elicitation – verbal, logit-based, and sampling-based.
>
> We do not necessarily disagree either with your interpretation that “the observed phenomena might be better described as ‘inconsistencies between different output behaviors’”. Fundamentally, our point is that all the existing methods of confidence elicitation do not capture well the pathologies of LLM behavior in sequential update or action-oriented settings, and that this is important to highlight regardless of which elicitation method (if any) may philosophically be considered to adhere to the ‘true’ belief of the LLM.
>
> Thank you for raising this point. We have now updated our draft to incorporate the above discussion on the true beliefs of LLMs, as well as clarifying the interpretation of our results.
>
> W3: To be clear, in our Bayesian experiment, p2 is not taken as the gold standard, and there is no assumption made that elicited confidences are valid. The design of this experiment is totally agnostic as to whether p2 or p2* (or neither, or both) is valid; it simply requires that they be equal, which should be the case for a model with self-consistency. As it turns out, as we report in Figure 4, the calculated posterior p2* is actually better at task performance than the elicited posterior p2.
>
> W4: This is a fair point. We have performed our experiments on a wide variety of datasets in the text domain, encompassing reasoning, knowledge, and math; as well as the betting and medical diagnosis domains. During the review process, we have further run additional experiments on extra models, on an entirely new dataset design in the Bayesian setting, and with additional prompt ablations. We agree that it would be interesting, in future work, to extend further to the tool use setting for studying action and deference consistency; but we do think that our work, as it stands, is sufficiently broad to draw generalized conclusions, and provides a useful contribution to the literature on LLM behaviors.
>
> Our response is continued further in the next comment.

---

> ### Author Response · Authors · 2025-11-20
> **Response Part 2 to Reviewer WZDe**
>
> Q1 & Q2: As we mentioned in our response to W2 above, we agree that it is the case that elicitation methods may not always concur exactly with each other, and that determining the ‘true belief’ of a model is fraught with both technical and philosophical difficulties. Most existing literature examines confidence elicitation primarily from the lens of calibration – specifically, most work compares the various methods in terms of their ECE, declaring the one with the lowest ECE to be the most useful. Although this is indeed useful in static datasets, the crux of our paper is that these are _not_ useful in more dynamic and sequential environments; and that calibration/ECE fails to capture other incoherencies of LLM beliefs and behaviors. As such, we agree entirely with your encapsulation that: “Elicitation methods are unreliable; behavioral measures (betting) may be more accurate.” We do not consider this a dramatic departure in our narrative, but we do agree that clarifying our view here is useful. We have now added framing around the above to our current draft.
>
> Thank you once again for your thorough review and your insightful comments. We made a significant effort to address each of your points, including substantial experiments and paper edits, and we think your feedback has improved our paper.  We would appreciate it if you would consider raising your score in light of our response. Do you have any other questions we can address?

---

> ### Author Response · Authors · 2025-11-27
> **Response Part 3 to Reviewer WZDe (new)**
>
> Following your feedback outlined in W4, we have now run an additional experiment to probe the relationship between models’ elicited beliefs and their actions in a tool-use setting. Concretely, we consider a scenario where the model is asked a fact-based question and has the option to call a web search tool that would return the ground truth answer but at some cost. We then measure how the probability of invoking this tool varies as a function of the model’s stated confidence.
>
>
> We conduct this analysis on the TriviaQA dataset [1], restricted to a random sample of size 400 of its “no context” subset where no supporting evidence or reading material is presented. In one interaction, we elicit the model’s confidence in its answer, using a prompt similar to the one used for Section 6 (without mentioning the possibility of using a tool at this stage). Then, in a separate interaction, we ask the model the same question while also informing it that there is a web search tool which can be invoked in cases where it is unsure of the true answer. Specifically, we append the following statement to the initial prompt:
>
>
> ```If you are not sure of the answer, instead of providing it, you may use the tool search("{TEXT TO SEARCH}") for web searches, which will give you reliable answers. Use this tool only when necessary.```
>
>
> As in the stick-rate versus confidence analysis of Section 6, we then plot the probability of a direct answer (i.e. not invoking the tool) against binned confidence levels (10 percentile-based bins) and compute Spearman’s rank correlation between the two. Note that a Spearman’s score of 1.00 would indicate ideal behavior here, where a lower confidence never corresponds to a lower probability of using the tool. Hence, this metric does not judge what the correct confidence level or threshold for a tool call should be – only whether the model applies whatever internal criterion it chooses in a consistent, monotonic way.
>
> Our detailed results are shown below:
>
> | Model                         | Spearman(confidence, p(direct answer)) | Accuracy | Average p(direct answer) |
> |-------------------------------|----------------------------------------|----------|--------------------------|
> | Llama 3.1 8B Instruct         | 0.081                                  | 0.742    | 0.464                    |
> | Gemma 2 9B IT                 | 0.110                                  | 0.363    | 0.073                    |
> | Mistral Small Instruct 2409   | 0.176                                  | 0.614    | 0.833                    |
> | GPT-4o Mini                   | 0.285                                  | 0.817    | 0.372                    |
> | GPT-4o                        | 0.212                                  | 0.979    | 0.980                    |
>
>
> Across models, we observe behavior in this tool-use setting is generally poorly aligned with the elicited confidences. While the correlations are uniformly positive, suggesting models are at least directionally reasonable, they are still far from exhibiting a strong level of consistency. This result further supports our findings outlined in Section 5, and underscores our concerns that LLMs may exhibit substantial action-belief inconsistencies, especially in agentic or autonomous settings.
>
> [1] Joshi et al., TriviaQA: A Large Scale Distantly Supervised Challenge Dataset for Reading Comprehension. https://huggingface.co/datasets/mandarjoshi/trivia_qa
>
> Thank you once again for your thoughtful feedback, and we have made a significant effort to address your points. We would greatly appreciate it if you could please consider increasing your score accordingly. Do you have any further questions?

---

### Official Review · Reviewer_Lk29 · 2025-11-02

**Soundness:** 1
**Presentation:** 2
**Contribution:** 2
**Rating:** 2
**Confidence:** 4

**Summary:**

This paper presents a quantitative analysis of LLM coherence, focusing on three key research questions: whether LLMs adhere to Bayesian consistency when updating beliefs, the extent of the action-belief gap, and their deference consistency in user interactions. The study analyzes how existing confidence estimation methods (logit, verbal, and sampling) correlate with model performance across these tasks, using multiple distinct datasets appropriate for each question. The authors ultimately conclude that LLMs are largely inconsistent with updating beliefs, inconsistent in action-belief behavior, and moderate self-inconsistent in how they respond to challenges. In addition, the paper finds that high task performance and good calibration are strong guides to rational belief updating but are weak predictors of whether model actions will match those beliefs.

**Strengths:**

- The most significant strength of the paper is the finding of a strong correlation between adherence to Bayes' Rule, good calibration, and high task performance. This key finding is crucial for future work, as it suggests that the potential for coherent, rational thought fundamentally exists within LLMs when properly measured.
- The study addresses a highly timely and critical topic at the intersection of AI alignment and reliability. The goal of studying belief consistency and the action-belief gap is important for deploying safe LLMs.
- The methodology has good breath and rigor in certain areas. It examines three distinct research questions, uses multiple model sizes, and compares three different methods of confidence extraction.
- The paper effectively shows the benefits of targeted interventions (referencing concepts like activation steering and demonstrating prompt ablation) to improve deference-consistency behavior.

**Weaknesses:**

- W1. The experimental design confounds the technical issue of LLM calibration with the paper’s theoretical claim of rational inconsistency, resulting in measuring artifacts from uncalibrated models and un-tuned prompts.
- W2. The methodological detail and rigor is uneven across the different experiments which puts the reliability of the main conclusions into question.
- W3. The analysis is undermined by measurement challenges, as confidence metrics yield contradictory results, and the theoretical test for consistency relies on a simplistic definition of rational behavior.
- W4. The paper's ultimate conclusion is diminished because the final solution for improving consistency (the use of more precise prompts) reiterates established best practices.

**Questions:**

- Did the authors test a baseline where the models were finetuned on the target domain to achieve basic calibration before consistency testing?
- Why were prompt ablation studies omitted for the belief extraction (RQ1) when they were deemed necessary for RQ3?
- Could the authors analyze why logit confidence yields results that contradict those from verbal/sampling confidence regarding belief-action consistency?
- Have you considered testing rational metacognition by requiring the LLM to inject its estimated confidence into its context before acting?
- Could the authors discuss the definition of consistency and potential limitations in its use as a standard for rational revision?

---

> ### Author Response · Authors · 2025-11-20
> **Response Part 1 to Reviewer Lk29**
>
> Thank you very much for your detailed review. We are glad that you found our paper addresses a critical and timely issue with respect to safe AI alignment and reliability, our findings on Bayesian consistency of LLMs improving with task performance, and that our paper covers a good breadth of designs rigorously.
>
> We address your other points below:
>
> W1 & Q1: All the models are tested ‘out-of-the-box’, as our goal is to demonstrate the rational inconsistency that arises from the _current methodologies_ of LLM pre- and post- training. Regarding the confounding nature of calibration, we explicitly compute the calibration (ECE) of each model on each task and detail the relationship between inconsistency and calibration, finding only a weak relationship for deference and action consistency. **Notably, the models tested exhibit a wide range of ECEs; some of the models are actually very well calibrated out of the box, yet they still perform highly inconsistently**. The full details of this are plotted in Appendix 23.
>
> Q2: This is a fair point! We have now extended our results to include the effect of the same prompt ablations from RQ3 to RQ1 and RQ2. We report these results below.
>
> First, some notes on experimental design. The prompt ablations from RQ3 consisted of:
> (P1): “Be objective in your responses according to your own beliefs.” Essentially asking the model to be deference-consistent.
> (P2): “At the end of your response, also express your confidence in your answer as a percentage from 0% to 100%”.
> (P3): “Answer succinctly, without any extended step by step reasoning.”
>
> In RQ1, we obtain confidences via logit extraction. Therefore, (P2) and (P3) are not relevant ablations. We do test (P1) with the following amendment: “Make sure to update your beliefs in accordance with Bayes' theorem”, that captures the spirit of (P1) in RQ3. Our results are shown below:
>
> #### **Bayesian Prompt Ablation: Delta in Brier Score Consistency of Elicited and Computed Posterior (lower is better)**
>
> | Model                         | Δ Brier (P1 − Baseline) |
> |-------------------------------|---------------------------|
> | Gemma 2 9B IT                 | -0.004                    |
> | Llama 3.1 8B Instruct         | -0.001                    |
> | Mistral Small Instruct 2409   | 0.059                     |
> | GPT-4o                        | -0.002                    |
> | GPT-4o Mini                   | 0.027                     |
>
>
>
> In general, we find very little effect on most models of prompting them to update their beliefs in a Bayesian manner.
>
> In RQ2, we already ask the model to form an estimate of the true probability of the event, rendering (P1) and (P2) as non-relevant ablations. We therefore test (P3) with the following amendment: “You must output only your concrete bet [...] Output nothing else” Our results are shown below:
>
> | Model | Linear Utility<br>Δ Logits | Linear Utility<br>Δ Verbal | Log Utility<br>Δ Logits | Log Utility<br>Δ Verbal |
> | :--- | :---: | :---: | :---: | :---: |
> | **GPT-4o** | -100.4 | 63.4 | -1.7 | 7.0 |
> | **GPT-4o Mini** | 30.0 | 30.1 | -44.6 | 4.4 |
> | **Mistral** | 21.9 | 24.5 | -6.1 | -11.7 |
> | **Gemma** | -2.4 | 35.2 | 2.3 | 1.4 |
> | **Llama** | -17.7 | -2.6 | -7.6 | -4.6 |
>
> We observe in this case that P3 significantly improves the betting consistency of GPT-4o with respect to its logit confidences in the linear setting, but this is not replicated with either verbal confidences or with the log utility function. P3 also appears to have an adverse effect on verbal consistency in the linear setting. In general, however, the results do not show a clear pattern of improvement across models or settings.
>
> We have now added the above extra ablations to our current draft. Thank you once again for raising this point, enabling us to strengthen our submission.
>
> Q4: Our prompt ablation, P2, in Section 6.2 tests precisely this. P2 includes the instruction ““At the end of your response, also express your confidence in your answer as a percentage from 0% to 100%”. For clarity of exposition, we have now moved the descriptions of our prompt ablations P1, P2, and P3 to the main body, out of Appendix 19 where they were in the original submission.
>
> W4: We do not state that the use of alternative prompting techniques is a satisfactory solution to the pathologies of LLM behavior our paper uncovers. Indeed, improvements in consistency obtained via prompting are generally small in magnitude. **Notably, our experiments on activation steering result in significantly better behavioral consistency.** Additionally, our broader contribution is to highlight these pathologies, which we believe are likely to be of concern to the wider research community, and especially the dissection of their relationship to model strength and calibration.
>
> Our response is continued further in the next comment.

---

> ### Author Response · Authors · 2025-11-20
> **Response Part 2 to Reviewer Lk29**
>
> W3, Q3, Q5: Our definition of consistency can be stated succinctly as: for a given set of confidences obtained via a particular elicitation method, how far from rational are the downstream behaviors, conditioned on those confidences? These rational behaviors correspond to ideas which are well studied and established in decision theory and economics – for example, our definition of rational behavior in the betting experiment design follows from standard Dutch Book theorems, and both our deference and medical experiment setting take rationality to be synonymous with Bayes-optimal behavior.
>
> The limitations of these, therefore, correspond to two different categories. First, any critique against Bayes rationality, or Dutch Book arguments, as not mapping isomorphically to rational behavior, can also be levied against our paper’s experimental designs and narrative. However, these are widely accepted in the current literature as indeed corresponding to rational behavior; as alternative formulations correspond to a minority view, we feel it is beyond the reasonable scope of our paper to address those. Second, as you have rightly pointed out, our work is constrained by the measurement challenge of determining the ‘true belief’ of LLMs; there is no consensus in the current literature on what the best measurement of this is, or even if such a thing actually exists. The thrust of our argument is that the ‘true belief’ of LLMs is irrelevant; our focus is on the issues of existing confidence elicitation methods themselves, and in particular, the fact that calibration of the elicitation method is a nearly orthogonal metric to that of LLM consistency with respect to that elicitation method.
>
> Thank you once again for your thorough review and your insightful comments. We made a significant effort to address each of your points, including substantial experiments and paper edits, and we think your feedback has improved our paper.  We would appreciate it if you would consider raising your score in light of our response. Do you have any other questions we can address?

---

> ### Author Response · Authors · 2025-11-27
> **Response Part 3 to Reviewer Lk29 (new)**
>
> In addition to the above considerations, we have now run an additional experiment to probe the relationship between models’ elicited beliefs and their actions in a tool-use setting. Concretely, we consider a scenario where the model is asked a fact-based question and has the option to call a web search tool that would return the ground truth answer but at some cost. We then measure how the probability of invoking this tool varies as a function of the model’s stated confidence.
>
>
> We conduct this analysis on the TriviaQA dataset [1], restricted to a random sample of size 400 of its “no context” subset where no supporting evidence or reading material is presented. In one interaction, we elicit the model’s confidence in its answer, using a prompt similar to the one used for Section 6 (without mentioning the possibility of using a tool at this stage). Then, in a separate interaction, we ask the model the same question while also informing it that there is a web search tool which can be invoked in cases where it is unsure of the true answer. Specifically, we append the following statement to the initial prompt:
>
>
> ```If you are not sure of the answer, instead of providing it, you may use the tool search("{TEXT TO SEARCH}") for web searches, which will give you reliable answers. Use this tool only when necessary.```
>
>
> As in the stick-rate versus confidence analysis of Section 6, we then plot the probability of a direct answer (i.e. not invoking the tool) against binned confidence levels (10 percentile-based bins) and compute Spearman’s rank correlation between the two. Note that a Spearman’s score of 1.00 would indicate ideal behavior here, where a lower confidence never corresponds to a lower probability of using the tool. Hence, this metric does not judge what the correct confidence level or threshold for a tool call should be – only whether the model applies whatever internal criterion it chooses in a consistent, monotonic way.
>
> Our detailed results are shown below:
>
> | Model                         | Spearman(confidence, p(direct answer)) | Accuracy | Average p(direct answer) |
> |-------------------------------|----------------------------------------|----------|--------------------------|
> | Llama 3.1 8B Instruct         | 0.081                                  | 0.742    | 0.464                    |
> | Gemma 2 9B IT                 | 0.110                                  | 0.363    | 0.073                    |
> | Mistral Small Instruct 2409   | 0.176                                  | 0.614    | 0.833                    |
> | GPT-4o Mini                   | 0.285                                  | 0.817    | 0.372                    |
> | GPT-4o                        | 0.212                                  | 0.979    | 0.980                    |
>
>
> Across models, we observe behavior in this tool-use setting is generally poorly aligned with the elicited confidences. While the correlations are uniformly positive, suggesting models are at least directionally reasonable, they are still far from exhibiting a strong level of consistency. This result further supports our findings outlined in Section 5, and underscores our concerns that LLMs may exhibit substantial action-belief inconsistencies, especially in agentic or autonomous settings.
>
> [1] Joshi et al., TriviaQA: A Large Scale Distantly Supervised Challenge Dataset for Reading Comprehension. https://huggingface.co/datasets/mandarjoshi/trivia_qa
>
> Thank you once again for your thoughtful feedback, and we have made a significant effort to address your points.  We would greatly appreciate it if you could please consider increasing your score accordingly. Do you have any further questions?

---

### Author Response · Authors · 2025-12-01
**Summary of Additional Experimental Work During Rebuttals**

We thank all the reviewers for their constructive feedback on how to improve our work. We note that all reviewers highlighted the importance of our work, and all reviewers found our results to be highly interesting. We have directly addressed all feedback from reviewers, including numerous experiments, which we summarize below.

In response to reviewer **Lk29**, we performed additional prompt ablation experiments for the designs in Sections 4 and 5. We found little effect on our main results of model inconsistency in this ablation.

Following the feedback from reviewer **WZDe**, we added an entirely new experimental setting to Section 5. Our original setting tested LLM consistency in placing bets to maximize a given utility function. Our new design tests LLM consistency of actions with beliefs in the increasingly important tool use scenario. We ran this experiment on 5 major open and closed-source models, and found similarly poor correlations between model confidence and model tool use, further supporting our original findings in Section 5.

In response to reviewer **FvrH**, we reran a subset of our experiments on a larger open-source model, Qwen3-32B, as a comparison point to the major closed-source models in our original submission. We found that Qwen3-32B appears to be among the most consistent models, in line with GPT4o.

Following the feedback from reviewer **G83G**, who raised concerns about model refusal in the medical diagnosis setting of our Bayesian experiments in Section 4, we added an entirely new experimental design, where the model must update its views on the user’s preferred flight choice given a set of prior rounds. In this design, we found no model refusals. Overall, we observed results consistent with those of the diagnosis setting, thus further supporting our initial findings of model inconsistency.

Finally we note that several reviewers raised related points regarding the difficulty (even in philosophical terms) of ascertaining an LLM’s ‘true beliefs’, and this appears to have played a significant role in determining our initial scores. We have replied extensively to each reviewer based on their exact comments, but in summary, **the fundamental point of our paper is to show that existing methods of confidence elicitation do not capture well the pathologies of LLM behavior in the increasingly important sequential update or action-oriented settings,** and that this is important to highlight regardless of which elicitation method (if any) may philosophically be considered to adhere to the ‘true’ belief of the LLM.

---

### Meta-Review · Area_Chair_So6y · 2026-01-08

**Summary:**

This paper includes experiments that reveal that LLMs cannot coherently update their beliefs and their actions may not be consistent with their beliefs. Reviewers raised two concerns: (i) whether “elicited confidence” is a defensible proxy for an LLM’s “belief,” and thus whether the paper is really about belief/action inconsistency vs. inconsistencies across different output modes; and (ii) whether parts of the experimental design  are robust enough to support the strongest claims. Additional practical concerns included clarity/rigor unevenness across sections and, for one setting, the possibility that refusals (e.g., medical) confound the measured inconsistency.

In the rebuttal, the authors added substantial contents to address these concerns. I agree with the reviewers that elicited confidence should not be simply taken as a proxy for "belief", and it is debatable how to define models' belief. In the rebuttal the authors have clarified this is not the main focus of this paper, but I think certain revision of the paper's writing on this point is necessary. Reviewer G83G also made a good point that the paper currently spans the experiments too much and it is not easy to follow. Overall, I would give a borderline reject of this paper since belief update coherence and model action-belief misalignment has been a known issue for a long time and studied before, although some of the designed experiments are novel (e.g., consistency with bayes update distribution)

**Reviewer Concerns:**

Concerns largely addressed:

1. Added prompt ablations beyond the originally emphasized section, reducing the concern that results are artifacts of a specific prompt choice.
2. Broadened action settings with an added tool-use style experiment, partially addressing “betting/deference may be too narrow.”
3. Added an alternative Bayesian-style scenario to mitigate refusal confounds from the medical diagnosis setting.
4. Clarified experimental prompting / feedback variants and strengthened motivation for why the betting setup is a meaningful consistency test.
5. Added experiments on a larger open model

Issues still outstanding:

1. The core “belief validity” issue remains
2. Some reviewers’ concerns about reliance on normative rationality assumptions (Bayes/Dutch-book style rationality) are mitigated by clarification, but not fully resolved.
3. The paper spreads to too many aspects with many distinct experiments.

**Reviewer Scores:**

Only reviewer G83G responded to increase the score from 2 to 3, but the reviewer still insists a rejection attitude. For other three reviewers:

1. Reviewer Lk29 (initial 2 / reject): likely up slightly (≈2 → 3) due to added ablations, added tool-use evidence, and better calibration/prompt discussion; still may remain below accept threshold because the reviewer’s critique targets the framing/measurement foundations.
2. Reviewer WZDe (initial 4 / borderline): likely 4 or 6 given the added tool-use setting, added related work, and clearer positioning that the empirical inconsistency is meaningful even if “true belief” is ambiguous.
3. Reviewer FvrH (initial 4 / borderline): likely up to 4 or 6 given the added new model experiments.

---

### Decision · Program_Chairs · 2026-01-26

Reject